# Efficient Delivery of *FMR1* across the Blood Brain Barrier Using AAVphp Construct in Adult *FMR1* KO Mice Suggests the Feasibility of Gene Therapy for Fragile X Syndrome

**DOI:** 10.3390/genes14020505

**Published:** 2023-02-16

**Authors:** Kathryn K. Chadman, Tatyana Adayev, Aishwarya Udayan, Rida Ahmed, Chun-Ling Dai, Jeffrey H. Goodman, Harry Meeker, Natalia Dolzhanskaya, Milen Velinov

**Affiliations:** 1NYS Institute for Basic Research in Developmental Disabilities, Staten Island, NY 10314, USA; 2Rutgers Robert Wood Johnson Medical School, New Brunswick, NJ 08901, USA; 3Macaulay Honors College at Hunter CUNY, New York, NY 10065, USA; 4Department of Physiology and Pharmacology, SUNY Downstate Health Sciences University, Brooklyn, NY 11203, USA

**Keywords:** fragile X syndrome, gene therapy, *FMR1*, FMRP

## Abstract

Background Fragile X syndrome (FXS) is the most common inherited cause of intellectual disability and autism. Gene therapy may offer an efficient method to ameliorate the symptoms of this disorder. Methods An AAVphp.eb-hSyn-mFMR1IOS7 vector and an empty control were injected into the tail vein of adult *Fmr1* knockout (KO) mice and wildtype (WT) controls. The KO mice were injected with 2 × 10^13^ vg/kg of the construct. The control KO and WT mice were injected with an empty vector. Four weeks following treatment, the animals underwent a battery of tests: open field, marble burying, rotarod, and fear conditioning. The mouse brains were studied for levels of the *Fmr1* product FMRP. Results: No significant levels of FMRP were found outside the CNS in the treated animals. The gene delivery was highly efficient, and it exceeded the control FMRP levels in all tested brain regions. There was also improved performance in the rotarod test and partial improvements in the other tests in the treated KO animals. Conclusion: These experiments demonstrate efficient, brain-specific delivery of *Fmr1* via peripheral administration in adult mice. The gene delivery led to partial alleviation of the *Fmr1* KO phenotypical behaviors. FMRP oversupply may explain why not all behaviors were significantly affected. Since AAV.php vectors are less efficient in humans than in the mice used in the current experiment, studies to determine the optimal dose using human-suitable vectors will be necessary to further demonstrate feasibility.

## 1. Introduction

Fragile X syndrome (FXS) is a neurodevelopmental disorder that is caused by an X-linked mutation in the *FMR1* gene. The overall prevalence of FXS is approximately 1 in 7000 males and 1 in 11,000 females [1,2]. In almost all known cases, the FXS phenotype is due to an expansion of more than 200 repeats and the subsequent methylation of CGG triplets in the 5′ untranslated promoter region of the *FMR1* gene [1,2]. The other FXS cases, in which triplet repeat expansions are not found, are often due to missense and nonsense mutations or deletions in the *FMR1* gene. All cases result in absent or markedly decreased production of the fragile X messenger ribonucleoprotein (FMRP) (formerly, fragile X mental retardation protein). The FMRP regulates the translation of approximately 4% of the fetal brain mRNA and directly regulates several classes of ion channels [3,4,5,6]. Clinically, patients with FXS present with distinct physical and behavioral features. Physically, these individuals have characteristic facial abnormalities (e.g., elongated face, large ears), large testes, hyperlaxity of joints, seizures, and hypotonia. Behaviorally, male patients with FXS typically have intellectual disability, anxiety, attention deficit hyperactivity disorder, poor eye contact, excessive shyness, hand flapping, hand biting, aggression, tactile defensiveness, hyperarousal to sensory stimuli, and autism spectrum disorder [7,8]. Female patients with FXS have a more variable phenotype.

The behavioral phenotype of males with FXS is hypothesized to be caused by an altered balance in excitatory and inhibitory neurotransmission and by the absence of the FMRP effect on synaptic plasticity and activity-dependent protein translation [9]. The routine management of FXS traditionally involves targeting the specific behavioral manifestations [8]. These therapeutic approaches have had variable success in controlling the FXS-associated deficits. No currently approved curative therapies exist, and clinical management continues to focus on symptomatic treatment of the comorbid behaviors and psychiatric problems [10].

The FMRP is involved in the regulation of multiple neuronal functions. Canonically, FMRP’s main function is that of an mRNA binding protein that regulates the localization, stabilization, and translation of approximately 4% of the fetal mRNA.

Many of the mRNA transcripts regulated by the FMRP are responsible for the production of synaptic proteins that are crucial for the high-fidelity information processing at the synapse. There are also non-canonical FMRP functions that include nuclear transcription regulation and protein–protein interactions [3,4,5,6,7]. Multiple clinical trials have aimed to address the specific functional deficits resulting from FMRP deficiency. However, to date, no definitive large-scale, placebo-controlled trials have met primary endpoints leading to drug use indications specific to FXS [11]. This is perhaps because no experimental drug that targets specific FMRP-associated neuronal dysfunction has led to comprehensive correction of the FMRP deficit [11,12].

Adeno-associated viruses (AAVs) are extensively used as vehicles for gene transfer to the nervous system. AAVs are well suited for these applications because they provide safe, long-term expression. Most of these applications rely on local AAV injections into the adult brain to bypass the blood–brain barrier (BBB) and to restrict transgene expression temporally and spatially [13]. Various recently engineered AAV9 serotypes have shown a higher efficiency for crossing the BBB, creating the possibility for the peripheral application of gene therapy constructs to treat CNS disorders [14]. Most notably, the AAV9 vector was successfully used to develop an *SMN1* gene construct that allowed peripheral administration for the treatment of Spinal Muscular Atrophy and was recently approved by the FDA for patient use [15].

The *FMR1* expression construct was previously administered directly into the brain via intraventricular and intracisternal magna administration to the *FMR1* knockout mouse and rat disease models [16]. The mouse and rat isoform homologs of human isoform 17 were used. The study showed an unequal distribution of FMRP after construct injection with relatively lower expression in the hippocampus, amygdala, and cerebellum. There was a partial behavioral rescue in the *Fmr1* KO rat model. These results indicate that the level of expression in some brain regions may need to fall within a narrow range for efficient attenuation of the behavioral phenotype [16].

In previous experiments, we observed very efficient expression of FMRP in a KO mouse model after peripheral injection of an AAV.php.eb vector construct that included the mouse isoform 7 and a human synapsin promoter. Robust FMRP expression was observed in all tested brain regions [17] with full poster data available at https://www.researchgate.net/publication/355788640_Efficient_Delivery_of_FMR1_across_the_blood_brain_barrier_using_AAVphp_construct-feasibility_of_gene_therapy_for_Fragile_X_syndrome, accessed on 1 February 2023. The current experiments will test if peripheral administration of the *FMR1* construct to adult WT and KO mice will lead to central expression of *Fmr1* and changes in the behaviors associated with FXS.

## 2. Methods

### 2.1. Research Animals

The mice used in this study were obtained from Jackson Laboratories (Bar Harbor, ME, USA) at six weeks of age +/− three days. The subjects included 24 FVB.129P2-Pde6b^+^ tyr^c-ch^ Fmr1^tm1Cgr^/J, *Fmr1* knockout (KO) and 12 FVB.129P2-Pde6b^+^ tyr^c-ch^/AntJ, wildtype (WT) male mice. The mice were housed in a vivarium at the New York State Institute for Basic Research in Developmental Disabilities (IBR). The animals were maintained with ad-lib access to chow and water under a 12:12 h light and dark cycle. All experiments were performed during the light phase from 9:00 AM to 5:00 PM. Oversight for the experiment was provided by the IBR Institutional Animal Care and Use Committee.

### 2.2. Drugs/Gene Therapy

ssAAVphp.eb-hSyn-mFMR1IOS7, a single-stranded AAV php.eb construct with the mouse *Fmr1* isoform 7 gene and human synapsin promoter, was purchased from Virovek Inc. (Hayward, CA, USA). The synapsin promoter was previously shown to be neuron specific [18]. Mouse isoform 7 corresponds to human isoform 17. This isoform was used because of its high level of brain expression and the observed partial phenotype correction in previous experiments using direct brain application [16,19]. FMRP ISO7 was reported to be the highest expressing transcript associated with polyribosomes in the adult mouse brain [19].

The delivery construct (2 × 10^13^ vector genome per kilogram, vg/kg) was diluted in sterile PBS and delivered in a final volume of 250 μL. The treatment was administered as a single dose by intravenous (IV) injection through the tail vein at eight weeks of age and four weeks prior to the first behavioral experiment (Figure 1). The treatment groups included (*n* = 12 mice per group) a WT control group that received an empty vector (WT + Empty), a KO control group that received an empty vector (KO + Empty), and a treatment group that received the viral vector (KO + Fmr1). There was one unexpected death in the KO + Fmr1 group 29 days following the treatment.

### 2.3. Procedures

All mice underwent a battery of four tests in the following order: open field, marble burying, rotarod, and fear conditioning. The first three tests were performed 24 h apart, while the last test was run during the following week. The mice were brought to the testing suite at least 1 h before any tests started. Before all tests for each subject, the apparatus was cleaned with 70% ethanol and water, and then dried with a paper towel, except the rotarod, which was cleaned with soap and water. The mice were euthanized with cervical dislocation 24 h after the last test.

### 2.4. Open Field

This 15-min test was performed in a 40 cm × 40 cm × 34.5 cm apparatus with transparent plexiglass walls [20]. The center of the apparatus was defined with a 7.5 cm × 7.5 cm square. The video tracking software ANY-maze (Stoelting, Inc., Woodale, IL, USA) was used to score the total distance traveled, average speed, and time spent in the center, while grooming and rearing behaviors were scored by hand.

### 2.5. Marble Burying

This 30-min test was conducted in a 25 cm × 25 cm × 14 cm apparatus with 5 cm deep bedding and 20 marbles arranged in a 4 by 5 grid placed approximately 2 inches apart [21]. After 30 min, the mice were returned to their home cage, and the number of marbles buried was manually recorded when a marble was more than half covered in the bedding.

### 2.6. Rotarod

This experiment was conducted in a motorized rotarod apparatus (AccuScan Instruments, Inc., Columbus, OH, USA) with a rotating rod divided into 4 testing chambers [21]. This test consisted of 3 trials, each 5 min apart with a rotation rate accelerating from 4 to 40 revolutions per minute (rpm) over 5 min. The software Fusion Rotarod 31 (AccuScan Instruments, Inc., Columbus, OH, USA) was used to track the duration of the activity, speed, and fall latency of each subject.

### 2.7. Contextual Fear Conditioning

This test was run over 2 consecutive days in a 20 cm × 20 cm × 27.5 cm plexiglass apparatus with a metal rod base [22] The ANY-maze software was used to record freezing behavior. Day 1 was the training session where the subject was first introduced to the conditioned (context) and unconditioned stimulus (1 mA foot shock, 2 s). Training involved a 2 min habituation period followed by the unconditioned stimulus. The mice were evaluated for one minute after the shock. Day 2 was the contextual fear conditioning trial in the same apparatus for 5 min without the unconditioned stimulus.

### 2.8. Mouse Tissue Lysate Preparation

The experimental animals were sacrificed with cervical dislocation, and the brains were removed. Following a longitudinal fissure cut, one half of the brain was subjected to structure-specific dissection (neocortex, hippocampus, cerebellum), weighed, and snap frozen on dry ice. Tissue lysates were prepared by mixing the roughly minced tissue in an ice-cold homogenizing buffer (20 mM Tris-HCl, pH 7.5, 150 mM NaCl, 1 mM EDTA, 0.5% Sodium deoxycholate, 1% NP-40, 0.1% SDS supplemented with Complete Mini Protease Inhibitor Cocktail (Roche Diagnostics Deutschland GmbH, Mannheim, Germany)) to achieve a 10% weight by volume solution. The lysates were homogenized in a prechilled Dounce homogenizer and further pulse-sonicated in an ice bath in a Branson digital signifier for 15 s (in cycles of 0.5 s on and 0.2 s off). The post-centrifugation supernatants (30 min at 16,000× *g* at 4 °C) were separated and stored at –70 °C. The tissue for the off-target screening for peripheral FMRP expression was processed similarly.

### 2.9. Mouse FMRP Quantification (qFMRPm)

The Luminex-based method was used for the quantification of FMRP as previously described [23]. In short, qFMRPm assays were prepared in 96-well format using monoclonal antibody mAb 5C2 coupled to xMAP Microplex microspheres (Luminex Corporation, Austin, TX, USA). The protein concentration of the tissue lysates was determined with a Pierce BCA Protein Assay Kit (Thermo Fisher Scientific, Waltham, MA, USA) and 3–10 mg of the total protein lysate/homogenate used per assay well in duplicates and repeated in three independent runs. The FMRP capture was carried out for 6 h in the dark in a 100 μL reaction volume (containing 50 μL of either diluted sample or GST-MR7 standard and 3000 mAb 5C2-coupled microspheres) and incubated at room temperature with agitation. For detection, anti-FMRP R477 rabbit polyclonal antibody diluted to 1.6 μg/mL in assay buffer was incubated at 4 °C overnight. Phycoerythrin-conjugated goat anti-rabbit IgG was used for fluorescent labeling of R477 for two hours at room temperature. The assay buffer was used for all inter-step washes and the resuspension of microspheres for reading in the Luminex-200 system (Luminex Corporation, Austin, TX, USA). The mean fluorescent intensity (MFI) was measured and analyzed with Luminex xPONENT software version 3.1 (Luminex Corporation, Austin, TX, USA). A coefficient of variance of 15% was set as the cut-off value for retesting.

### 2.10. Statistical Analysis

Statistical (estore.onthehub.com) was used to perform one-way or repeated measures factorial ANOVAs with Fisher’s LSD post-hoc test for all experimental data and graphs.

## 3. Results

### 3.1. Efficiency of FMRP Delivery and Off-Target Distribution

Neuron-specific targeted delivery of the *Fmr1* isoform 7 was selected as the most age appropriate for the time of the therapy administration. The level of FMRP detected in the WT + Empty treatment group was comparable with that previously observed per the structure profile of the FMRP expression in the FVB mouse strain [23]. Efficient delivery was observed in all tested CNS tissues in the treated KO mice (KO + Fmr1). The FMRP expression in different tested brain regions is shown in Figure 2. The FMRP levels in the KO + Fmr1 mice were generally higher than in the WT controls and varied by brain region. Consistent with previous findings [13], the vector delivered greater transduction in the cortical structures and was noted to have slightly reduced transduction efficiency in the cerebellum (Figure 2D).

No off-target delivery of FMRP was observed in the various tested tissues in the KO + Empty control mice and the treated KO + Fmr1 mice (Figure 3). This indicates efficient targeted delivery to the brain and may decrease the side effects associated with FMRP expression outside the brain.

### 3.2. Behavioral Tests

#### 3.2.1. Open Field

Figure 4 shows the exploratory behaviors, anxiety-like behaviors, and locomotor activity from the open field test [16,24]. The gene therapy had no effect on the distance traveled (F_2,32_ = 1.58, NS), average speed (F_2,32_ = 1.54, NS), time spent grooming (F_2,32_ = 2.00, NS), number of rears (F_2,32_ = 2.71, NS), and time spent in the center (F_2,32_ = 0.65, NS). This suggests that the exogenous expression of FMRP has no effect on locomotor activity and anxiety-like behavior.

#### 3.2.2. Marble Burying

Figure 5 shows repetitive digging behavior evaluated with the marble burying test. The number of marbles buried was not significantly affected by the gene therapy (F_2,32_ = 1.03, NS). This suggests that the exogenous expression of FMRP has minimal or no effect on repetitive behavior.

#### 3.2.3. Rotarod

Figure 6A shows motor coordination and learning evaluated with the accelerating rotarod test. The mice showed motor learning by remaining on the rotarod longer across the trials (F_2,32_ = 7.09, *p* < 0.01); specifically, the mice stayed on the rotarod longer during trial 3 than during trials 1 (*p* < 0.001) and 2 (*p* < 0.05). There was a gene therapy by trial interaction (F_4,64_ = 7.09, *p* < 0.01). The WT + Empty and KO + Fmr1 mice stayed on the rotarod longer each trial, while the KO + Empty remained on the rotarod the same through all three trials. WT + Empty mice remained on the rotarod longer in trial 3 than trial 1 (*p* < 0.01) but not trial 2 (NS). The KO + Fmr1 mice stayed on the rotarod longer during trial 3 than trial 1 (*p* < 0.001) and trial 2 (*p* < 0.01). However, the KO + Empty mice performed better during the first trial than the WT + Empty mice (*p* < 0.05) and the KO + Fmr1 mice (*p* < 0.05).

The KO + Fmr1 mice demonstrated motor learning in the rotarod test compared to the KO + Empty mice. The KO + Empty group performance on trial 1 is consistent with the higher locomotion of the group observed in the open field. However, they failed to exhibit any progress on consecutive trials, lacking the learning process. In contrast, the rotarod performance of the WT + Empty and KO + FMR1 mice showed progressive learning with each consecutive trial. Based on this, correlations between the rotarod performance and FMRP levels in the hippocampus and neocortex of the individual mice were performed. The result of these correlations is shown in Figure 6B–D. There was an inverse correlation between the performance on the rotarod test and the FMRP levels, suggesting that FMRP oversupply may interfere with the attenuation of the phenotype.

#### 3.2.4. Contextual Fear Conditioning

Associative learning and memory were assessed by freezing behavior during the training and the contextual trials (Figure 7A,B). There was an effect of training as the mice froze longer after the training than during habituation (F_1,32_= 33.05, *p* < 0.001; Figure 7A). The gene therapy had an effect on training (F_2,32_= 8.48, *p* < 0.01), as the mice with *Fmr1* expression, the WT + Empty mice (*p* < 0.001), and KO + Fmr1 mice (*p* < 0.05) froze more after the unconditioned stimulus. The WT + Empty mice froze significantly more than the KO + Fmr1 mice (*p* < 0.001). The reduced freezing following the unconditioned stimulus by the KO + Empty mice suggests that these mice were less affected by the unconditioned stimulus, leading to reduced learning.

The context trial examined the strength of the association between the shock and context by measuring the time spent freezing when returned to the training context 24 h later (Figure 7B). The context test was analyzed across the 5-min test. The WT + Empty mice froze more overall than the other two groups (F_2,32_ = 10.03; *p* < 0.001). There was a trend towards an interaction of group and time (F_4,128_ = 2.32; *p* = 0.06), which revealed that during the first minute of the context test, the KO + Fmr1 mice froze significantly more than the KO + Empty group (*p* < 0.05). The KO + Fmr1 mice were not significantly different from the WT + Empty mice during minutes 1 and 5 of the context test. This suggests that the gene therapy had a mild effect towards rescuing learning and memory.

## 4. Discussion

Our study shows that a gene therapy construct injected into the peripheral bloodstream of adult mice can generate efficient FMRP expression in the central nervous system. As a proof of concept, the vector was administered to adult mice in which the synaptic connections were expected to be fully mature, instead of during development when the brain is more plastic. Administration of the gene therapy construct to adult mice was not expected to show effects as robust as those of treatments administered during earlier stages of development.

These data are intended to move forward the development of non-invasive gene therapy for FXS. There was efficient delivery of *Fmr1* in all tested brain regions and minimal off-target delivery. This was made possible by using a neuron-specific promoter in the delivery construct. CNS specific delivery is expected to decrease the possible side effects of the therapy associated with off-site *FMR1* expression. The endogenous FMRP expression is not limited to neurons. Although FMRP is always expressed in neurons, the glial FMRP is under developmental regulation and normally decreases in the adult mouse brain [25]. Since this study was designed for adult animals, we specifically focused only on neuron-targeted gene therapy. The behavioral tests in this study were conducted to evaluate the effect of Fmr1 gene therapy administered to adult mice in the environment of established neuronal connections, on motor learning, repetitive behaviors, and learning and memory. The administration of the gene therapy construct led to the Fmr1 knockout mice performing similar to the WT mice in the accelerating rotarod, where motor learning across the trials was evaluated. In addition, the gene therapy led to partial correction of learning and memory in the contextual fear conditioning experiment. In all tests performed, the learning and memory components were the parts most affected by *Fmr1* re-introduction. In a study by Hooper et al., [16] Fmr1 KO rats injected with AAV with human *FMR1* isoforms exhibited only mild effects in the open field test and contextual fear conditioning, similar to the findings in the current study. When gene therapy was introduced at P0 in Fmr1 knockout mice and their behavior was evaluated at 6 weeks of age, there was no effect on locomotor activity and learning and memory, but gene therapy reduced anxiety-like behavior and repetitive behaviors [26]. Marble burying was reduced in mice when the therapy was administered at P0 but not at 8 weeks of age, suggesting that the repetitive behavior becomes resistant to intervention with age in the mice.

Fmr1 KO mice exhibit repetitive digging behavior by burying more marbles, which was reduced by agmatine [27] and MPEP [28]. In the current study, the KO + Empty mice did not bury more marbles than the WT + Empty. Because of this, it is unclear if the empty vector led to an increase in the number of marbles buried by the WT mice or reduced the marbles buried by the KO mice. Therefore, it is unclear if the gene construct had an effect on the KO mice, as these mice buried a similar number of marbles.

Previous studies showed similar results to the current findings where the empty vector did not change the hyperactivity, and higher anxiety-like behavior was shown by less time in the center of the open field exhibited by the Fmr1 KO mice [16]. The gene construct did not affect these behaviors.

Individual mouse correlations between FMRP levels and rotarod performance showed an inverse correlation of the performance with FMRP levels. This suggests that the overexpression of FMRP in some brain regions may be associated with decreased efficiency of correction of the disease phenotype. In support of this conclusion, *FMR1* duplications in humans were previously associated with phenotypes of cognitive disability [29,30].

The AAV vector used in our experiments, AAVphp.eb, was previously shown to have over 50 times higher efficiency in crossing the BBB when used in C57BL/6J mice. Using such a vector made it possible, for the first time, to achieve sufficient neuron specific FMR1 expression after peripheral application. However, the use of this vector has its limitations because its high BBB permeability is facilitated by interactions with the LY6A (SCA-1) protein that is only expressed in the brain microvascular endothelial cells of some mice strains, including the parental mouse strain FVB/6J used in our study [31,32,33]. Therefore, this vector is not expected to have the same efficiency in humans. On the other hand, only approximately 20% of normal levels of FMRP have been shown to be sufficient to restore neuronal functioning in vitro [34]. Moreover, mosaicism for methylation of *FMR1* is shown to be associated with higher functioning in individuals with FXS [35]. Accordingly, to further suggest the feasibility of this approach for human therapy, the following experiments should test the possibility of ameliorating the FXS phenotype using AAV vectors applicable in humans and with careful dosing of the viral genome.

This study has the following limitations. The FMRP levels in specific brain cells were not determined. This may be important to address in future studies since cell specific *FMR1* expression may be of importance for the efficiency of phenotype correction. In addition, in our experiments, FMRP was not detected outside the brain likely due to the used neuron-specific promoter. This makes it difficult to monitor the *FMR1* expression in the brain using noninvasive methods for the purpose of human gene therapy. Additional tissues to look for *FMR1* expression that were not included in our study and may be considered in future experiments include peripheral blood and testes. Alternatively, downstream markers of *FMR1* expression may be investigated.

## Figures and Tables

**Figure 1 genes-14-00505-f001:**
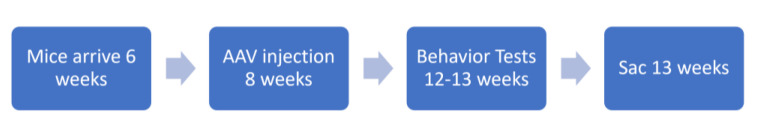
Timeline of the experimental procedures.

**Figure 2 genes-14-00505-f002:**
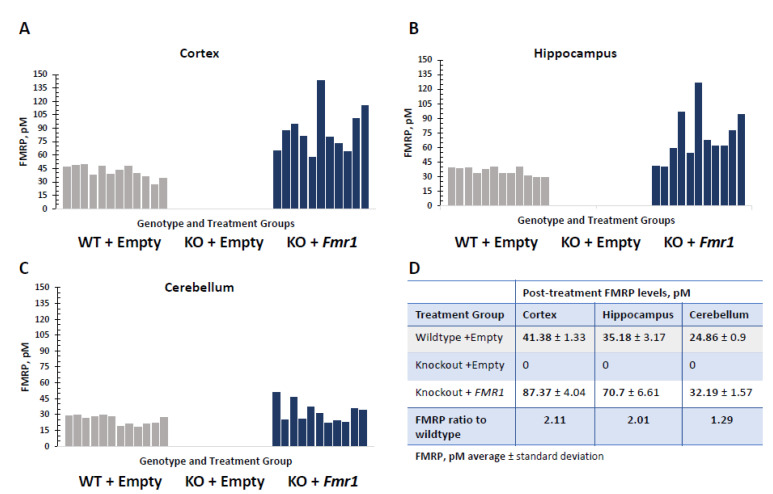
FMRP expression in different brain regions in *Fmr1*-treated KO mice (KO + Fmr1) compared to wildtype (WT + Empty) or KO (KO + Empty) mice that were treated with an empty vector. The FMRP levels in the cortex (**A**), hippocampus (**B**), and cerebellum (**C**), where each bar represents individual animals of each treatment group, are shown in the corresponding panels. Wildtypes (WT-) are shown in gray, and *Fmr1*-treated knockout (KO+) are shaded blue. The average FMRP expression per treatment group in the mouse brain structures is shown in the table (**D**).

**Figure 3 genes-14-00505-f003:**
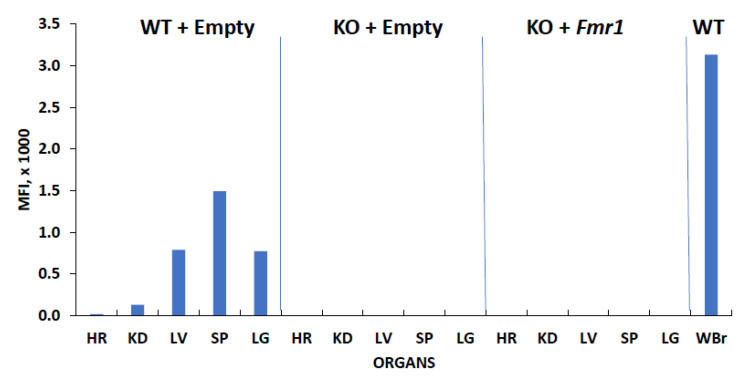
FMRP expression in various tissues. The mouse tissue lysates (10 μg) from the wildtype (WT + Empty) and knockout animals with (KO + Fmr1) and without (KO + Empty) *Fmr1* treatment were subjected to relative qFMRPm assessment. The FMRP levels in the whole brain in the WT mice were used as the positive control for expression (WBr). Abbreviations of lysates: heart (HR), kidney (KD), spleen (SP), liver (LV), lung (LG).

**Figure 4 genes-14-00505-f004:**
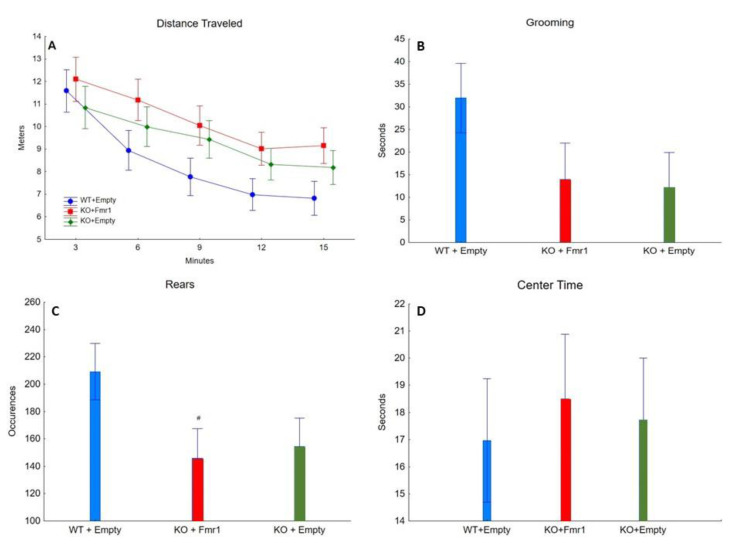
Open Field. The effect of the three treatments on open field. (**A**) Distance traveled (in meters) by the mice from the three groups during the 15 min of open field. (**B**) The time (seconds) the mice of the three groups spent grooming. (**C**) The number of times the mice of the three groups engaged in rearing. # = *p* < 0.05. (**D**) The time (seconds) the mice of the three groups spent in the center of the apparatus throughout the test.

**Figure 5 genes-14-00505-f005:**
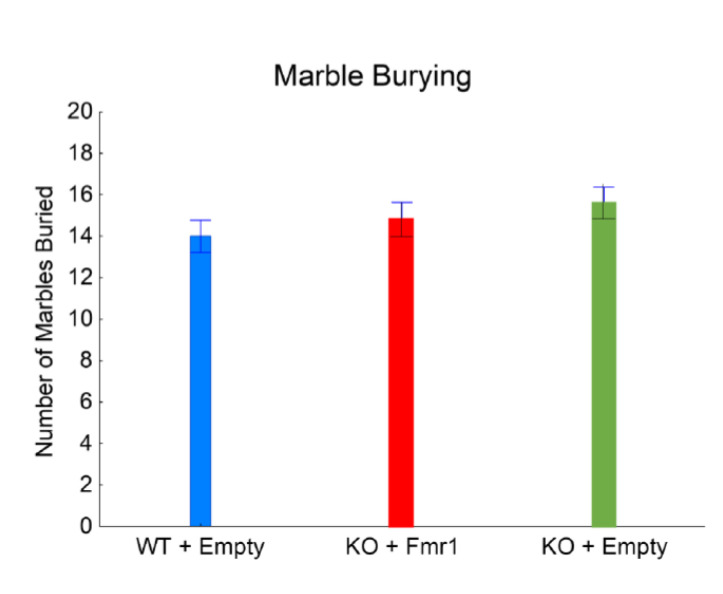
Marble burying. The effect of the gene therapy on the number of marbles buried.

**Figure 6 genes-14-00505-f006:**
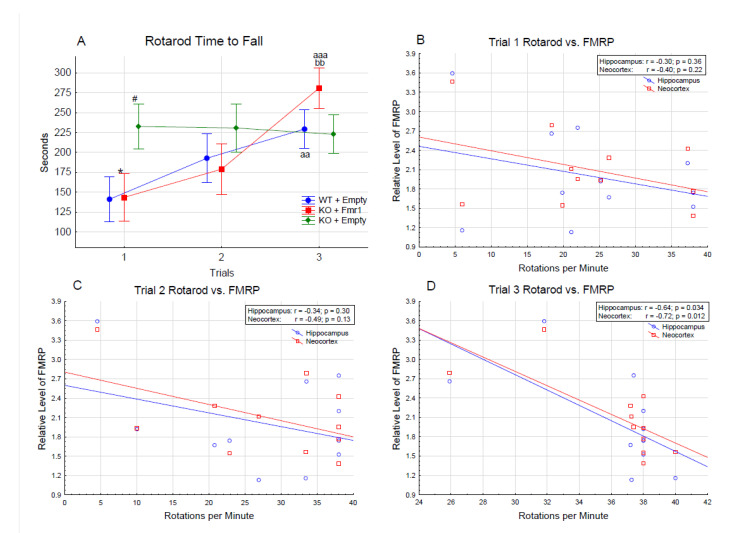
(**A**) Accelerating rotarod. The difference in the time to fall (in seconds) between the treatment groups over the 3 trials of the rotarod test. Differences from the WT + Empty group are shown by # = *p* < 0.05. Differences from the KO + Empty group are shown by * = *p* < 0.05. Differences from the first trial are shown by aa = *p* < 0.01, and aaa = *p* < 0.001. Differences from the second trial are shown by bb = *p* < 0.01. Correlations of rotarod performance with FMRP levels in hippocampus and neocortex of individual mice in trial 1—(**B**), trial 2—(**C**), and trial 3—(**D**).

**Figure 7 genes-14-00505-f007:**
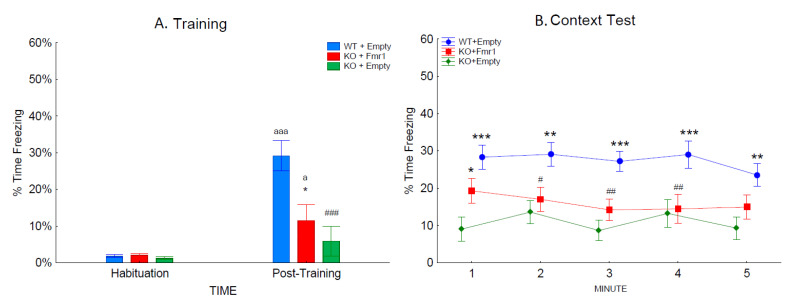
Contextual fear conditioning. The effects of the treatments on associative learning during training and contextual fear conditioning. (**A**) The percentage of time the mice of the three groups froze during habituation and the post-training period during training. (**B**) The percentage of time the mice of the three groups froze during re-exposure to the training context. Differences from the WT + Empty group are shown by # = *p* < 0.05, ## = *p* < 0.1, and ### = *p* < 0.001. Differences from the KO + Empty group are shown by * = *p* < 0.05, ** = *p* < 0.01, and *** = *p* < 0.001. Differences from habituation are shown by a = *p* < 0.05, and aaa = *p* < 0.001.

## Data Availability

Data from this study may be provided if requested.

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
