# Peer review of "Efficient Delivery of FMR1 across the Blood Brain Barrier Using AAVphp Construct in Adult FMR1 KO Mice Suggests the Feasibility of Gene Therapy for Fragile X Syndrome"

_genes, 2023, doi:10.3390/genes14020505_

Round 1
Reviewer 1 Report
The manuscript describes an important study in development of gene therapy for the Fragile X Syndrome (FXS), the most common inherited form of intellectual disability in humans. It is based on an animal model of FXS. However, the step from mice or rats to humans is huge, so that conclusions should be drawn very carefully.

Author Response
Reviewer 1
I suggest following minor corrections or improvements:
- In Title the word "demonstrates" could be replaced by "suggests" to be more cautious in interpretation of the results. Mice are not humans!
The change has been made as suggested.
- In the abstract, line 15, it should be stated at which age/how many days after treatment the mice were put to perform the various behavioral tests. The names of the various tests may be mentioned in the main text.
The following edits were made:
Line 14 “…tail vein of adult Fmr1 knock…”
Line 15 has been modified to read “Four weeks following treatment, …”
- In the Introduction, lines 40-45, there is unnecessary repetition: “Behaviorally FXS males...” and “The behavioral phenotype of FXS males...” These sentences should be united and shortened to one sentence.
Lines 41-53 have been modified to read “Behaviorally, male FXS patients typically have intellectual disability, anxiety, attention deficit hyperactivity disorder, poor eye contact, excessive shyness, hand flapping, hand biting, aggression, tactile defensiveness, hyperarousal to sensory stimuli, and autism spectrum disorder[7,8]. Female patients have a more variable phenotype
- Introduction, line 59, would benefit from a new paragraph or subtitles in the whole introduction.
New Paragraph was created
- Introduction, lines 83-88. The authors report their unpublished results (ref. 22 is an abstract). One would like to have here a published reference, or if not possible, add the words "unpublished data” instead of ref. 22.
The data reported on this poster presentation is fully available on ResearchGate. The link to the entire poster was included in reference 22.
- Pages 3-4, The Methods part lacks references. Are all the methods novel and unpublished, except ref. 23 of quantitation of FMRP? The results of the study cannot be replicated without more details on methods, possibly as supplementary material. See comment 2 on treatment time as an example. The selection of the behavioral tests is most interesting. What do these tests really measure, are they relevant for human behavior point of view and do the results depend on age or sex? Have these been taken into account?
References in support of the use of synapsin promoter and the isoform 7 were added in section 2.2. References have been also added for the behavioral tests (sections 2.4, 2.5, 2.6, 2.7).
- I would suggest to add a time-line of treatment to test as a schematic figure.
A timeline of experiments has been added (Figure 1).
- Results p. 5. It is questionable whether individual animal data need to be visualized like this. Could you use group mean or median values + SD or SE and make just one diagram? This data exists, in fact, in D. In D, however, the treatments are abbreviated differently from figures A-C. I think WT – is a bit misleading and so is KO +, they are not as clear as just WT and KO + Fmr1.
New modified Figure 1 (now as Figure 2) was added.
- Page 6 Fig. 2. Among the tissues studied there was not testis lysate. FMRP is expressed in the testis in WT mice. Why is it missing?
Testis lysate was not collected, and such data is missing. We acknowledge this omission and will correct this in future experiments.
- Page 7 in Fig. 3A and 3B are identical curves! The y-axis is different, but distance travelled (A) is a function of speed (B) so one figure is enough.
Answer: figure 3B was removed.
- Page 8 lines 262-264 belong to Methods.
This line reports how the KO+Empty mice did during training, so we believe it should remain in the results section.
- Discussion p. 10, lines 309-314. Can you draw conclusions at all, or what do you think about the marble burying test in this case?
A sentence has been added: “Marble burying was reduced in mice when the therapy was administered at P0, but not at 8 weeks of age, suggesting that the repetitive behavior becomes resistant to intervention with age in the mice.”
- Discussion p. 10, lines 318-319: The sentence can be improved, e.g. “Rotarod performance showed individual inverse correlation between…”
The sentence has been re-written “Previous studies showed similar results to the current findings where the empty vector did not change the hyperactivity and higher anxiety-like behavior as shown by less time in the center of the open field exhibited by the Fmr1 KO mice [21].”
- Discussion p. 10, line 332. As in the Title, replace “demonstrates” with “suggests”. There is such a huge gap in understanding differences in brain functions of mice and humans.
The change has been made as suggested.
- References: editing needed. Ref. 22 to be replaced or removed.
As mentioned in answer 5, the data reported on this poster presentation is fully available on ResearchGate. The link to the entire poster was included in reference 22.
Reviewer 2 Report
In this manuscript, the authors showed an efficient, brain-specific delivery method of Fmr1 via peripheral administration in adult mice. They used ssAAVphp.eb-hSyn-mFMR1IOS7, a single-stranded AAV php.eb construct with the mouse Fmr1 Isoform 7 gene and human synapsin promoter to treat the Fmr1 KO mice .The gene delivery led to increased FMRP level and partial alleviation of the Fmr1 KO phenotypical behaviors. Overall, the study provides new information to explore an AAV.php.eb vector -based gene therapies against Fmr1 KO phenotypical behaviours in mice . This research subject would be relevant to the reader's interest in Genes .However, several potential issues need be addressed.
#1 The authors used human synapsin promoter to induce expression of mFMR1 in AAVphp.eb-hSyn-mFMR1IOS7 vector. Did author observe only mice neurons are highly infected ? There are many kinds of cells in the brain, such as neurons, astrocytes and microglia, etc. , which cell type is most susceptible for this AAV infection in the brain must be clarified . In additional FRMP levels presented in the figure 1, it is encouraged that authors quantify efficiency of AAV- mFMR1IOS7 infection in different brain regions shown by immune staining too. In other studies , adeno-associated virus 2 (AAV2) with CMV promoter is also highly expressed human CNS including retinal cells ( i.e., RPE cells PMID: 32755565) . it is valuable to include this information in discussion to elaborate advantages and limitations to use AAVphp.eb-hSyn-mFMR1 is this study .
#2 Page 4, line 180: FMRP gene has many isoforms, why chose Fmr1 isoform 7 as the delivery target? Whether inappropriate isoform resulted in the negative results of neuroethology?
#3Figure 5A: In the accelerating rotarod test, the authors showed that the WT+Empty and KO+Fmr1 mice remained on the rotarod shorter in trial 1. It's confusing that WT mice showed poor motor learning ability even though in trial 1.
#4
AAV gene therapy may produce hepatorenal toxicity, and we should also pay attention to whether high doses of AAV virus cause other side effects. A simple biochemical test of mouse blood is recommended.
#5
Page 1, line 30 in “Introduction”: “Almost all known cases” should probably be changed to “In almost all known
Author Response
Reviewer 2
- The authors used human synapsin promoter to induce expression of mFMR1 in AAVphp.eb-hSyn-mFMR1IOS7 vector. Did author observe only mice neurons are highly infected ? There are many kinds of cells in the brain, such as neurons, astrocytes and microglia, etc. , which cell type is most susceptible for this AAV infection in the brain must be clarified . In additional FRMP levels presented in the figure 1, it is encouraged that authors quantify efficiency of AAV- mFMR1IOS7 infection in different brain regions shown by immune staining too. In other studies , adeno-associated virus 2 (AAV2)with CMV promoter is also highly expressed human CNS including retinal cells ( i.e., RPE cells PMID: 32755565) . it is valuable to include this information in discussion to elaborate advantages and limitations to use AAVphp.eb-hSyn-mFMR1 is this study.
Our experiments did not include separate measurements of FMRP levels in neurons and other CNS cells. We agree that it would be of interest to explore these issues in future experiments. Immune staining in different brain region in one treated mouse was done in our initial experiments as referenced in reference 22. A link to the entire poster data was added to this reference. Such immune staining was not done in the current study because of logistic constraints.
- Page 4, line 180: FMRP gene has many isoforms, why chose Fmr1 isoform 7 as the delivery target? Whether inappropriate isoform resulted in the negative results of neuroethology?
Isoform 7 of the mouse Fmr1 corresponds to human isoform 17. This isoform is the second most highly expressed in the brain. As reported in reference 21, this isoform was used in previous experiment via direct injection and showed efficient partial phenotype correction. Additionally, FMRP ISO7 was reported to be highest expressing transcript associating with polyribosomes in adult mouse brain. Citation regarding this was included in 2.2. These were the reasons to use this isoform. Additional text clarifying the rationale for using this isoform was added to section 2.2.
.
- Figure 5A: In the accelerating rotarod test, the authors showed that the WT+Empty and KO+Fmr1 mice remained on the rotarod shorter in trial 1. It's confusing that WT mice showed poor motor learning ability even though in trial 1.
The KO+Empty group performance on trial 1 is consistent with the higher locomotion of the group seen in the Open Field. However, they failed to exhibit any progress on consecutive trials, lacking the learning process. In contrast, the rotarod performance of WT+Empty and KO+FMR1 showed progressive learning with each consecutive trial.
- AAV gene therapy may produce hepatorenal toxicity, and we should also pay attention to whether high doses of AAV virus cause other side effects. A simple biochemical test of mouse blood is recommended.
While we have not done such measurements, this will be considered in future experiments.
- Page 1, line 30 in “Introduction”: “Almost all known cases” should probably be changed to “In almost all known.
The change has been made as suggested.
Round 2
Reviewer 2 Report
The following is my comments. minor revision.
"The revised manuscript is greatly improved . To provide unbiased information to readers , I suggest to add brief answers in the discussion part about reviewer 1 raised question one including previous reports of CMV-AAV2 expression in CNS ".
Author Response
"The revised manuscript is greatly improved . To provide unbiased information to readers , I suggest to add brief answers in the discussion part about reviewer 1 raised question one including previous reports of CMV-AAV2 expression in CNS ".
Answer: text was added to the discussion to address this comment, please see lines 336-340.